# Production-induced seismicity indicates a low risk of strong earthquakes in the Groningen gas field

Nepomuk Boitz [1] ✉, Cornelius Langenbruch[1] & Serge A. Shapiro [1]

The maximum possible earthquake related to gas production in Western Europe's largest gas field, Groningen, Netherlands, is an urgent practical question. Here we show how to distinguish between induced and triggered tectonic earthquakes. We estimate the maximum possible induced magnitude in the Groningen gas field to be around $M_w = 4$. We extend the concept of the seismogenic index to gas-production, and calculate the worst-case probability of triggering a larger-magnitude tectonic earthquake in a continuum. The probability of a $M_w 5.5$ earthquake at Groningen is significantly higher than at Pohang Geothermal System (South Korea), where a $M_w 5.5$ earthquake was actually triggered. Due to a long history of production in Groningen, our model estimates that strong earthquakes ($M_w \geq 4$) must have occurred there several times, in disagreement with the observations. This indicates that the Groningen gas field is inherently stable and the physical conditions to trigger large tectonic earthquakes likely do not exist.

It has been known for decades that both subsurface fluid extractions and injections can cause earthquakes[1–6]. The question of the maximum possible earthquake magnitude, $M_{max}$, in connection with such geotechnical operations is important for understanding, evaluating and controlling their seismic hazard[7–9]. This question is under-researched and controversial, especially when it comes to the long-term production of hydrocarbons such as in the Groningen gas field[10].

The Groningen gas field is the largest in western Europe with a total gas volume of approximately 2900 bcm of gas which is located in the sedimentary formation Upper Rotliegend at approximately 3km depth. The reservoir exhibits an increasing thickness from south-east (150 m) to north-west (close to 300 m) (Fig. 1b) The field was discovered in 1959, and production started in 1965 and increased steadily until the 1970s, when more than 80 bcm of gas were produced annually. Afterwards, production declined to annual rates between 30 and 50 bcm per year (Fig. 1a).

After the largest earthquake ($M_w$ 3.5, $M_L 3.6$)[11,12] in 2012 lower production rates were mandated in the Groningen field. In response to production, pore pressure in the field decreased up to 1 MPa per year, with a total pressure depletion of approximately 28 MPa until 2022 (Fig. 1a). The end of gas production from the Groningen field is

planned. Only parts of the field will be kept open as a backup with a full closure planned for 2030.

The reason for the closure is the seismicity caused by gas production. In 1991, the first earthquake was recorded in the Groningen area, followed by a rather constant number of earthquakes (around 10–20 annually above $M_c = 1.2$) until 2002. In the following years, an increasing rate of seismicity was observed (Fig. 1c). Until February 2022, altogether 1474 earthquakes with magnitudes between $M_L = -0.8$ and $M_L = 3.6$ were registered in the Groningen gas field (Fig. 1d). Since seismicity is expected to continue after the cessation of extraction, the question of the maximum possible magnitude for the field is still intensively and controversially discussed[13–18]. Moreover, case studies around the world show that maximum induced earthquake magnitudes sometimes occur after the termination of energy projects[19], which makes such an estimate for Groningen even more important.

In order to understand the nature of a maximum possible earthquake, a distinction must be made between earthquakes that are, on the one hand, predominantly caused by geotechnical interventions in the subsoil, and, on the other hand, tectonically prepared earthquakes that are triggered by anthropogenic influences[20–23].

[1]Earth Science Department, Freie Universität Berlin, 12249, Malteserstr. 74-100, Berlin, Germany. ✉e-mail: boitz@geophysik.fu-berlin.de

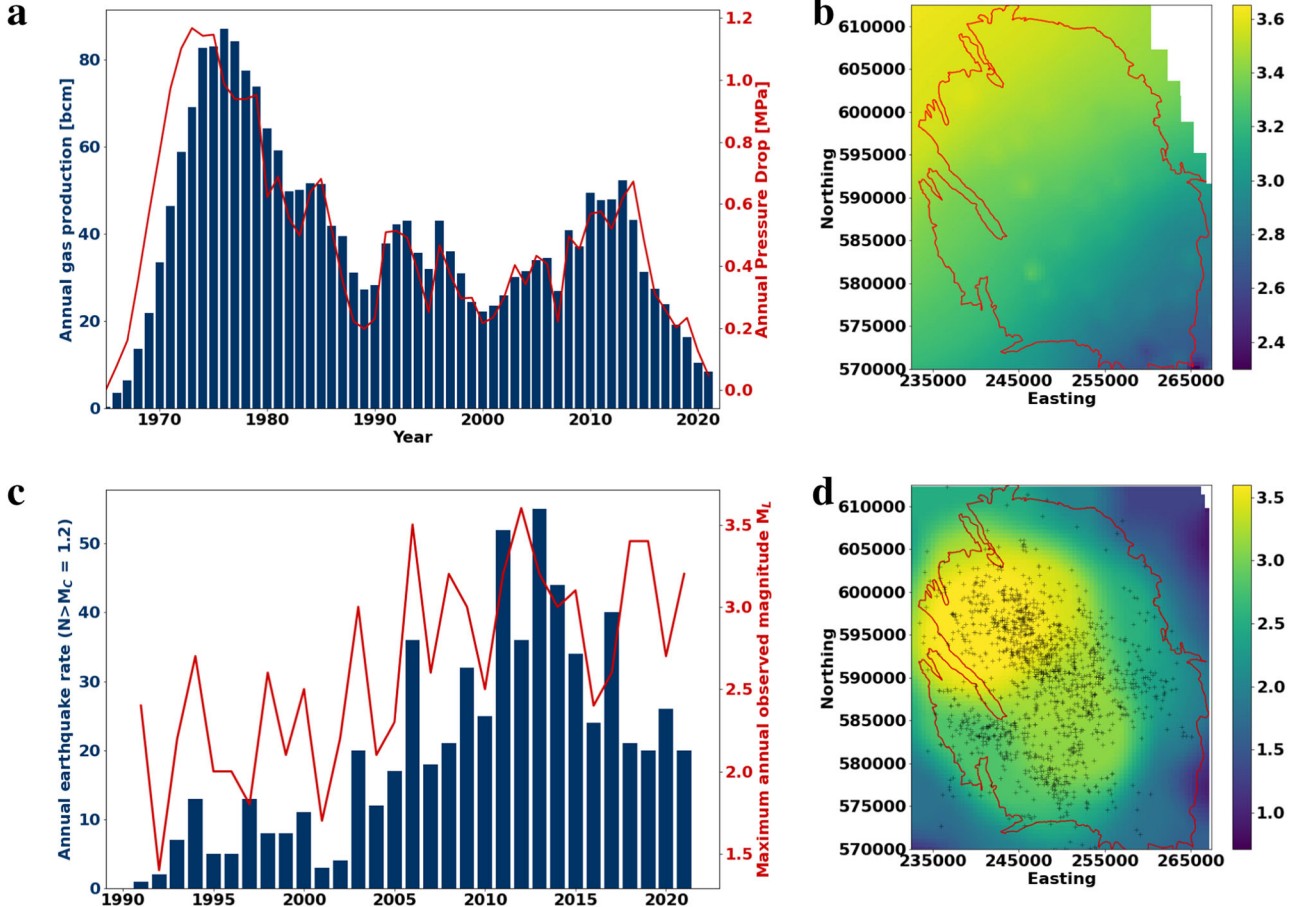

**Fig. 1 | Production rates and observed seismicity in the Groningen gas field.** **a** Annual production rates and annual pressure decrease averaged over all wells, **b** Moment magnitude calculated from Eq. (1) using the known local reservoir thickness and assuming a stress drop of 10 MPa. **c** Annual number of earthquakes above $M_C$=1.2 and maximum observed magnitude for each year, **d** Spatial distribution of observed earthquakes and maximum observed magnitudes within the Groningen field. Observed maximum magnitudes increase from SE to NW which is in good agreement with the theoretical estimates of reservoir-limited rupture sizes (1b).

Induced and triggered earthquakes are defined differently by different authors. One of the earliest approaches distinguishes between induced and triggered seismicity based on the stress impact of the stimulation[20]. If the shear stress release of an earthquake is in the order of this stress impact, the event is considered induced. If the shear stress release from the earthquake is significantly greater than the effect of the stimulation, the event is considered triggered. Another approach[2] considers the events as induced earthquakes, which only occur due to geotechnical operations and would not occur without them. They are fully controlled by man-made stress changes. In contrast, triggered events occur on faults that are favorably oriented relative to tectonic stresses. The fluid operations affect only the nucleation of such events, not the event magnitude or extent of the rupture. The third approach[21] defines induced seismicity as earthquakes occurring entirely within the stimulated rock volume, that is, their entire rupture surfaces lie within this volume. In contrast, the rupture surfaces of triggered earthquakes run outside the stimulated volume. It is clear that all three approaches are related. However, they emphasize different physical aspects of the same phenomenon. Because of a clear relation to the observable features of earthquakes, we use the third approach[21] of defining induced and triggered seismicity. Moreover, in respect to the triggered event, this approach is close to the more recently proposed concept of a runaway earthquake[22,24].

Induced seismicity is controlled mainly by technical operations and depends less on tectonic features. Thus, it is easier to control than triggered seismicity by modifying operation activity based on the application of traffic-light systems[25–27]. However, understanding both the maximum possible induced earthquake and the maximum possible triggered earthquake is vital for assessing the seismic hazard related to geotechnical operations, particularly in the Groningen gas field. The question arises whether it is possible to use induced seismicity to obtain indications of the possible triggering of large tectonic earthquakes. This question is of crucial importance for the extraction of geo-energy from the underground, which also applies to the Groningen gas field.

Here, we develop an approach to estimate the maximum possible induced magnitude $M_Y$ for the case of fluid extraction from a flat underground reservoir. The method is based on considering induced seismicity[21] as earthquakes occurring entirely within the reservoir layer stimulated by production. The frequency-magnitude distribution for this situation can be described by the so-called Lower-Bound frequency-magnitude statistic[21,28] (LB-statistic, see Methods). The LB-statistic is characterized by a shape like the truncated Gutenberg-Richter law, that is, larger magnitudes are underrepresented and an upper magnitude limit exists. We apply our approach to the Groningen reservoir and obtain a maximum possible induced magnitude of $M_Y$ = 4, a magnitude at the lower limit of commonly made estimates[13]. In the next step, we assume that triggering of larger tectonic earthquakes at Groningen is possible. We find that due to the long production history of the Groningen gas field the probability of triggering tectonic

earthquakes up to $M_w 5.5$ would be significant (up to 30%) in this case. Events of $M_w \geq 4$ should have happened there already several times, if possible. These results are based on the application of the Seismogenic Index (SI)[29,30] and the worst-case probability of triggered large tectonic earthquakes[31] described in the Method section. Our results suggest that the Groningen gas field is inherently stable and likely the physical conditions for triggering larger tectonic earthquakes do not exist.

## Results

### Theoretical magnitude estimates from reservoir geometry

For analysis of the Groningen seismicity, we initially assume that all earthquakes are induced, that is, the rupture size is limited to the reservoir layer stimulated by production. Moreover, we assume approximately isometric (e.g., nearly round) ruptures. Since the reservoir geometry is known, we can compute the maximum magnitudes for reservoir-limited ruptures[32]

$$M_Y = \log_{10}\left(X_Y^2\right) + \log_{10}(\Delta\sigma)/1.5 - 6.07, \quad (1)$$

where $X_Y$ is the effective rupture length and the $\Delta\sigma$ is the stress drop (SI units are used). The Groningen gas field is under normal-faulting conditions and we assume that ruptures are dipping 60° to the horizontal direction, in good agreement with dips derived from moment tensor solutions[33,34]. The largest possible rupture length under the assumed conditions is given by:

$$X_Y = \frac{h}{\sin(\pi/3)}, \quad (2)$$

where $h$ is the reservoir thickness.

We use stress drops of 1 and 10 MPa, which are approximately limiting the stress drops of significant events in Groningen[35–37]. Further, we assume that maximum rupture surfaces of induced earthquakes are limited by the reservoir thickness plus the fault offset at the given location. The resulting maximum rupture lengths $X_Y$ are in the order of 200 m in the SE and 500 m in the NW of the field. Corresponding maximum possible magnitudes are given by $2.6 \leq M_{max} \leq 3.3$ in the SE and $3.5 \leq M_{max} \leq 4.1$ in the NW, considering local reservoir thickness and stress drops in the range from 1 to 10 MPa. Our derived maximum magnitudes computed throughout the field (Fig. 1b) are in good agreement with observed maximum magnitudes (Fig. 1d). An increase of the maximum magnitude from the SE to the NW is actually observed. It is substantiated by a statistical analysis of the event-size distribution of the Groningen extraction-induced seismicity catalogue[14] concluding that the probability of larger magnitude events in the NW-region is statistically significantly larger than in the southern and eastern parts of the gas field. In addition, the majority of observed precise hypocenter locations of seismicity in Groningen are indeed restricted to the reservoir layer[33,34,38].

Our analysis explains these observations by considering ruptures limited to the reservoir layer perturbed by production. It suggests that all earthquakes in the Groningen field can be characterized as induced earthquakes. In the case of exclusively induced earthquakes, the frequency-magnitude distribution can be described by the so called Lower-Bound frequency-magnitude statistic[21,28], which we apply in the following section.

### Lower-Bound frequency-magnitude statistic of induced seismicity

The Lower-Bound frequency-magnitude statistic[21,28,39] (henceforth LB-statistic) assumes that the stimulated rock volume (approximately given by the cloud of hypocenters of the seismicity related to a geo-technological operation) represents a spatially-limited domain of an abstract statistically homogeneous infinite continuum (further the seismo-tectonic continuum) with seismo-tectonic properties of the

stimulated geologic formation. Conventionally, the Gutenberg-Richter (GR)[40] frequency-magnitude statistic is assumed in the seismo-tectonic continuum. However, the LB-statistic assumes that in this continuum only those earthquakes will be induced, the rupture surfaces of which are entirely located inside the stimulated volume.

We approximate the stimulated volume of the Groningen gas reservoir by a flat layer of local thickness. Based on observed normal-faulting conditions and fault dips of approximately 60°[33,34], we consider earthquakes occurring on rupture surfaces critically oriented with respect to the tectonic stress. Figure 2a shows a sketch of our application of the LB-concept to a single stimulated horizontal layer. Blue lines indicate possible, and red lines impossible rupture surfaces according to the LB-statistic. The dashed blue line indicates the largest possible rupture surface. Using these limiting conditions for possible rupture sizes, we derive explicit equations of the Lower-Bound frequency-magnitude statistics of induced seismicity (see Methods, Eqs. 4 and 5).

In contrast to the classical GR-statistic our formulation of the LB-statistic for induced seismicity includes an upper limit on the maximum possible magnitude $M_Y$, like the truncated GR-statistic[14,17]. However, Muntendam-Bos and Grobbe[14] show how notoriously difficult it is to constrain a magnitude bound or taper of the GR-statistic from a purely statistical analysis of observed seismicity at Groningen. We do not rely solely on the statistics of observed seismicity but include information about reservoir and rupture geometries in the analysis to determine if an upper magnitude bound exists. Thus, the upper magnitude limit and the functional form of the LB-statistics (see Methods, Eqs. 4 and 5) are physically substantiated and meaningful (see also Methods). This is in contrast to frequently applied approaches with artificially introduced magnitude tapering functions or magnitude truncation values in the GR-statistics[14,17].

Another useful feature of our LB-statistic is its independence of any (potentially unknown) geometric parameter of the stimulated volume, as all LB-statistic parameters can be obtained by fitting Eq. (4) to observed seismicity. Thus, our main geometrical assumptions are not significantly restrictive and the LB-statistic can be generally applied to seismicity observations to search for indications of the finiteness of seismically active domains.

Figure 2c shows non-linear LB-statistic fits of the exact solution (Eqs. 4) and an approximation (Eqs. 5, see Methods) to the observed frequency-magnitude (FM)-distribution at Groningen. In addition, we present the log-linear fit of the classical GR-statistic (Eq. 3). The determined fitting parameters (a, b, and $M_Y$) and their uncertainty are given in Fig. 2c. The exact LB-statistic solution (Eq. (4)) and the approximation (Eq. (5)) yield very similar results for a, b and $M_Y$ with a slightly higher uncertainty for the exact solution (see the Supplementary Information). The maximum possible magnitude according to the LB-statistic for the whole field (complete catalog) is $M_Y = 3.97$.

In addition to determining a possible maximum magnitude ($M_Y$) in the finite perturbed reservoir, a fitting of the LB-statistic provides us with the possibility to derive parameters of the seismo-tectonic continuum (e.g. b-values). The LB-statistic-based estimate of the b-value is smaller than the b-value derived from the classical GR-statistic (see Fig. 2c). This is explained by the fact that the GR-statistic does not account for any potential upper magnitude limit. If present, such a constraint in the data influences the GR-statistic fitting by biasing the b-value towards higher estimates. Thus, the smaller b-value estimated from the LB-statistic likely describes the magnitude scaling in the seismo-tectonic continuum, including ruptures running out of the stimulated volume.

However, we find that even the biased classical GR-fit unlikely describes the observed maximum magnitude. Based on a- and b-values of the classical GR-fit (b = 0.94, a = 3.96, Fig. 2c) and the Poisson assumption[29,41] (independent magnitudes) we compute the statistically expected maximum magnitude[42] given by $\hat{M}_{max} = \frac{1}{b}a = 4.21$, a value

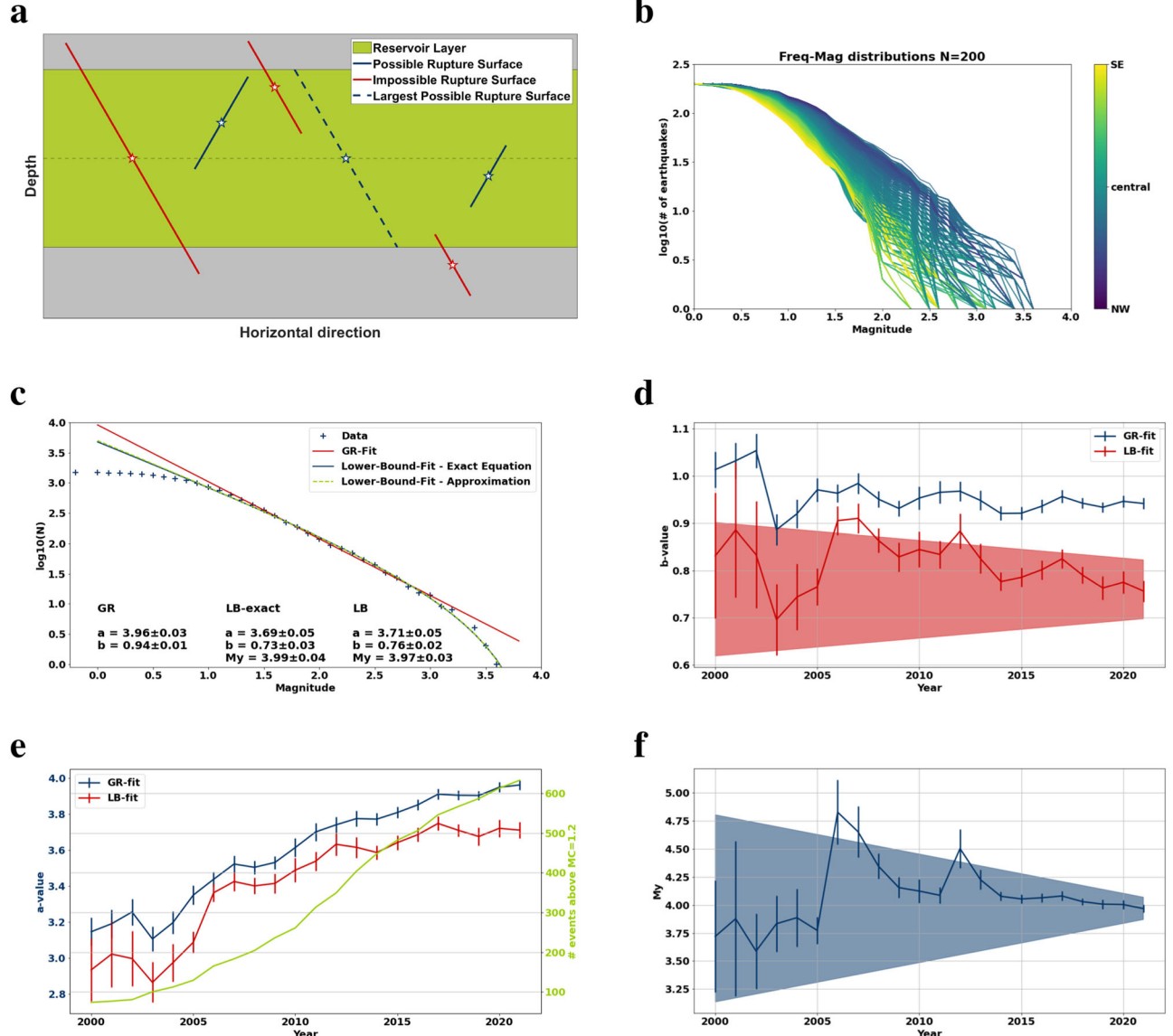

**Fig. 2 | Lower-Bound (LB) model and observed and fitted frequency-magnitude (FM) distributions. a** Sketch of the LB-model. Blue lines indicate possible, red lines impossible rupture surfaces, the dashed blue is maximum possible rupture surface. **b** Observed frequency magnitude distributions in Groningen, color-coded by spatial occurrence. The magnitude distribution shows a clear increase of larger magnitude events towards the NW **c** GR and LB fits of the total observed seismicity in Groningen, temporal distribution of **d** observed seismicity and the a-values, **e** the b-values and **f** $M_Y$, with uncertainties (error bars) obtained from covariance matrices of the fits. The reddish and bluish shaded areas show uncertainties obtained from the statistical analysis in the supplementary material. Uncertainties obtained from the covariance matrices seem to underestimate the statistical error, b-value and $M_Y$ can be assumed to be time-invariant.

significantly larger than the observed maximum magnitude of M=3.6. According to the fitted classical GR-statistic the probability to observe no event of magnitude larger than M=3.6 in the complete catalog is as low as 2.34% (see Supplementary Fig. S4). In other words, corresponding to the classical GR-statistic the probability to observe an event larger M = 3.6 is above 97%. Note also that the computed statistically expected maximum magnitude of $\hat{M}_{max} = 4.21$ is not an upper limit but the expectation based on the classical GR-fit. It will increase with increasing number of events in the catalog (the a-value) if seismicity at Groningen continues and the classic GR-statistic is valid.

We used the Akaike Information Criterion[43] (AIC) to assess if the classical GR-model or the LB-model is the preferred model to describe the observed frequency-magnitude distribution. The AIC penalizes additional parameters in a considered model to prevent over-fitting. While the fit of the LB-statistic obviously can explain the observed frequency-magnitude distribution more accurately (especially the systematic deviation for larger magnitudes) it has an additional free

parameter ($M_Y$). Nevertheless, the LB-model is characterized by a lower AIC (lower information loss) in the magnitude range $M \geq 1.5$ and thus it is the preferred model (see the Supplementary Information). It is again an indication that observed seismicity in Groningen is exclusively induced and an upper magnitude limit at $M = 4$ likely exists.

## Temporal distribution of seismicity in Groningen

In the last section, we applied the LB-statistic to the complete seismicity catalog in Groningen and compared it to the classical GR-statistic. Here, we estimated the temporal evolution of the parameters of the seismo-tectonic continuum and of the stimulated volume. As discussed above the b-value of the LB-statistic likely describe magnitude scaling in the seismo-tectonic continuum as classic GR b-values can be biased by existing upper magnitude limits. The b-value as well as $M_Y$ show temporal variations (Fig. 2d–f). The direct fitting of the classical GR-statistic to the observed frequency-magnitude distribution provides b-values that are relatively stable and close to 0.95 (Fig. 2d). The

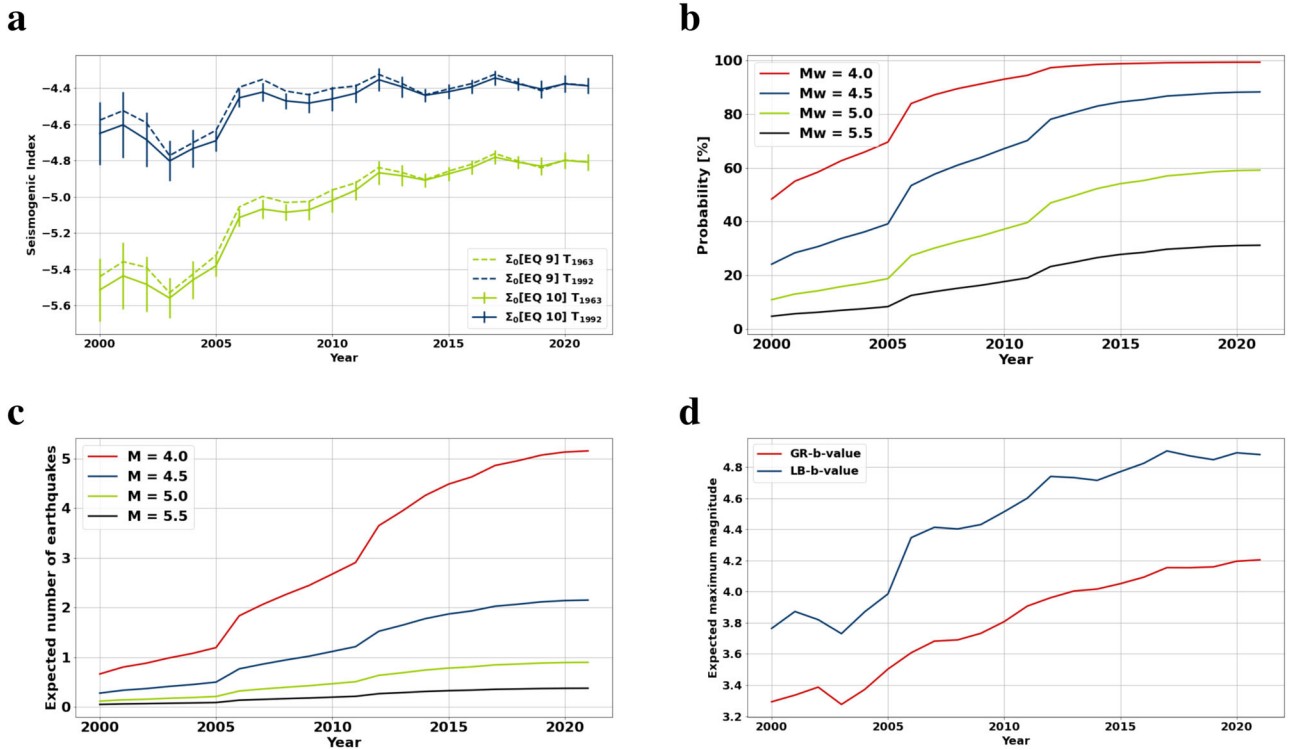

**Fig. 3 | Temporal evolution of the Seismogenic Index and the Worst Case Exceedance Probability (WCEP). a** Seismogenic indices estimated using Eqs. (9) and (10) for the total time period since the start of production and for the period since the first observed earthquake (1992). Error bars correspond to the uncertainties of a and b values, see Fig. 2. **b** WCEP of magnitudes M4, M4.5, M5 and M5.5 (see Methods for details). **c** Expected number of earthquakes $(-\ln(1 - WCEP/100))$ of the WCEP in **b**. The probability to trigger larger magnitude events as well as their expected number is high **d** Statistically expected maximum magnitudes[42] assuming b-values from the LB and GR-fits. The expected magnitudes are significantly larger than observed (see also Fig. S4).

LB-statistics estimates of the b-value in Groningen are initially small ($\approx 0.85$) and further decrease until 2003. The first larger magnitude event ($M_L = 3.5$) in 2006 causes an abrupt increase of the b-value to a value of 0.9. An increase in the b-value caused by a large magnitude event might seem contradictory but can be explained by the presence of aftershocks of the large event. The high level of statistical errors (see Fig. 2d) shows that the temporal variations of the b-value are mainly of statistical character and occur due to the limited number of observations in the first years of the earthquake sequence (see discussion in the Supplementary Information).

The temporal distributions of the b-value and the maximum possible magnitude $M_Y$ derived from the LB-statistic show similar features. The two larger-magnitude events in 2006 and 2012 caused an increase in both the maximum possible magnitude and its uncertainty. In the subsequent years these estimates decreased again (Fig. 2f). Note that all temporal estimates of the maximum possible magnitude $M_Y$ are larger than the observed $M_{max} = 3.6$.

Uncertainties in $M_Y$ and the b-value (Fig. 2f and d) are obtained from the covariance matrices of the corresponding non-linear fits and decrease with time, as observation statistics improve. Presented uncertainties might underestimate the actual errors, as we elaborate in the Supplementary Information. Overall, temporal changes of $M_Y$ and the b-value are likely of statistical nature. Thus, b-values of 0.94 (GR) and 0.76 (LB) should be used to reduce uncertainties in statistical analyses, because these values have been obtained from the complete catalog.

In the next step, we estimate the Seismogenic Index (SI) and its temporal evolution (Eq. (10), see Methods). The Seismogenic Index (SI)[29,30,44] quantifies a normalized seismic response of the seismo-tectonic continuum to a perturbation of the Coulomb Failure Stress (CFS). As discussed in the Methods section, the change of the CFS is controlled by the reported pore pressure change caused by

production (see Fig. 1). To account for a potentially incomplete earthquake catalog in the first years of the observation (the observation system before 1992 was sparse and observations are likely incomplete) and a possibly existing minimum triggering threshold of the CFS change, we assume two different scenarios to compute the SI.

In scenario 1, the induced seismicity is directly controlled by pore pressure changes since the start of production. In scenario 2, an initializing pore pressure perturbation is required for starting induced seismicity and it becomes controlled by pore-pressure changes after the first observed earthquake in 1992. Considering these two scenarios helps to constrain the real value range of the SI. In the case of scenario 1, the SI increases significantly in the time period between 2000 and 2012 and afterwards remains on a constant level (green line in Fig. 3a). In scenario 2, the SI is almost constant throughout the whole observation period (blue line in Fig. 3a). This would correspond to the SI behavior at a given field site with stationary conditions. Accounting for possibly missing seismicity observations before 1992 would move the estimates corresponding to scenario 1 closer to the estimates of scenario 2. Thus we obtain the SI in the range between the estimates of scenarios 1 and 2. The slight increase of the SI in scenario 2 with time can be explained by a gradual involvement of some cohesive faults in the seismic activation due to the pressure depletion and corresponding occurring earthquakes with an enhanced stress drop[31].

We also estimated spatial variations of the parameters of the seismo-tectonic continuum and the stimulated volume (see the Methods section for details). The estimated a-values (Fig. 4a) within the Groningen field are rather constant which can be explained by (1) the large smoothing distance of 15 km radius (see Methods section) and (2) a good interconnection (i.e. a large permeability) of the reservoir itself.

The spatial distribution of the b-value shows a clear segmentation into the NW part of the field with small b-values around 0.8 and the SE

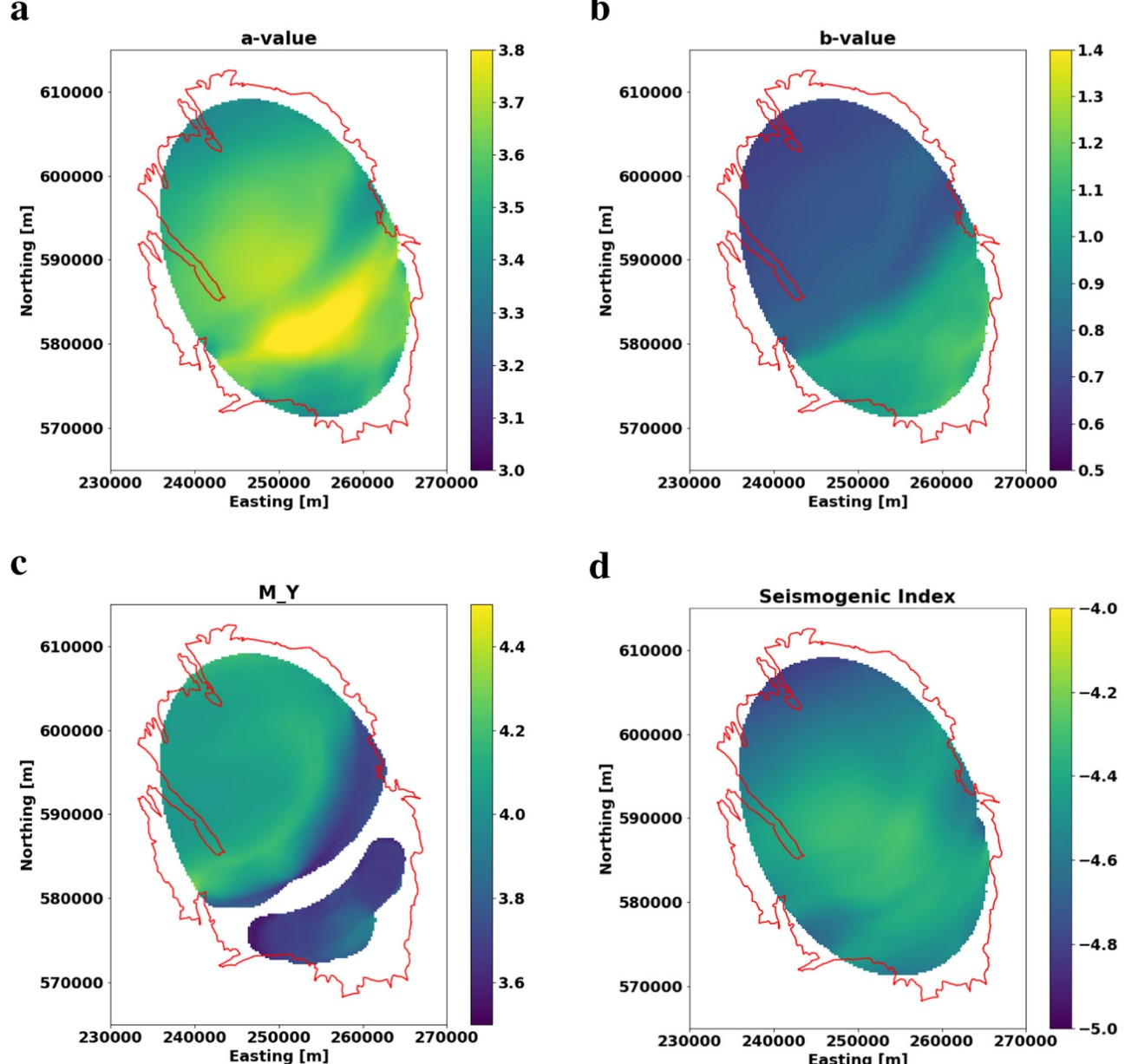

**Fig. 4 | Spatial distribution of the Lower-Bound (LB) parameters. a** the LB a-value, **b** the LB b-value, **c** $M_Y$ and **d** the Seismogenic Index. b-values and $M_Y$ differ significantly between SE and NW-part of the field. $M_Y$ is larger and the b-value smaller in the thicker parts of the reservoir, compare Fig. 1b. The a-value is high in the central part of the field and decreases towards the rim of the reservoir, the seismogenic index shows no significant spatial variations.

with b-values significantly larger than one (Fig. 4b). Although the b-value map is smoothed over circles with radii of 15 km, this segmentation is significant and agrees with results of a previous study[14]. The segmentation of the field into NW and SE parts is also evident in the $M_Y$ distribution. The NW parts exhibit $M_Y$ around 4.1, and the SE part smaller magnitudes around 3.7 (Fig. 4c). Note that at no location the observed maximum magnitude exceeds our estimate obtained from fitting the LB-statistic to the local FM-distribution.

Since the event density in the SE is lower, the uncertainty of $M_Y$ is larger. Here, we only plot $M_Y$ up to an uncertainty of 0.2 (see discussion in the Methods section) which causes empty space in the $M_Y$ map. Overall, the observed seismicity as well as the $M_Y$ estimates show the theoretically expected correlation with the reservoir thickness. Observed magnitudes as well as possible maximum magnitudes ($M_Y$) are significantly larger in regions where the reservoir is thicker.

The SI (Fig. 4d) shows no significant spatial variation. This is in agreement with the spatial behavior of the a-value and the pore-pressure depletion which are almost constant throughout the field, and superimpose the effect of the different reservoir thicknesses in SE and NW.

**The Worst-Case Exceedance Probability (WCEP)**
Finally, we use our estimates of the parameters of the seismo-tectonic continuum (the SI and the b-value) to estimate the probability of triggering a large-magnitude event in Groningen. In other words, we address the possibility to activate ruptures propagating beyond the stimulated volume. For example, this could correspond to activating faults extending into the basement. We attempt to find constraints of the probability of triggering large earthquakes in the seismo-tectonic continuum. Precisely, we compute constraints for the Worst-Case Exceedance Probability (WCEP) using Eq. (8), (see Methods and Fig. 3b). The

WCEP is computed using the upper bound of the SI and the lower bound of the b-value over time (i.e., the supremum of the SI and the infimum of the b-value over time). The WCEP for the two scenarios used to compute the SI are the same as the specific assumptions used in the computation of the SI are also used to obtain the WCEP.

We find that the WCEP to trigger magnitudes above $M_W > 4.0$ is high : we estimate a probability of ≈60% to trigger a $M_W = 5.0$ until 2022 (Fig. 3b). This is significantly larger than for example the WCEP of an $M_W 5.5$ event for the Pohang case study or an $M_W 3.4$ event for the Basel-EGS[31] which were both estimated to be around 15% and both in fact occurred. Even if we accept larger b-values provided by the fitting of the GR statistic, we obtain very significant values of WCEP of earthquakes with $M_W > 4$ (see the Supplementary Information).

Moreover, the WCEP probability (assuming a Poisson distribution) for events with $M_W > 4.0$ (the estimated maximum possible induced magnitude according to the LB-statistic) is about 99%. This implies that $M_W \geq 4.0$ events are expected to have happened already $-\ln(1 - WCEP/100) = 5.1$ times (i.e., more than 5 times) during the production period. In other words, the production period is five times longer than the recurrence intervals of such earthquakes. The expected number of earthquakes according to the WCEP is shown for various magnitudes in Fig. 3c. Because of the length of the production time interval, events of magnitude up to about M = 4.9 are expected in the case of ruptures propagating outside of the stimulated volume. However, these events have not happened to date.

Another possibility to estimate the statistically expected $M_{max}$ was proposed by van der Elst[42] using the b-value, SI, and injected fluid volume ($V$) for injection-induced seismicity: $\hat{M}_{max} = \frac{1}{b}(\Sigma + log_{10} V)$. Here, we apply the same concept by replacing the logarithm of the injected volume with $\delta\Sigma(t)$ (see Eqs. 7 and 9, Methods section). This yields the maximum magnitudes shown in Fig. 3d for parameters obtained from fitting the LB-statistic and the classical GR-statistic. By using the b-value from the LB-concept we obtain expected magnitudes of approximately 4.9 (or 4.21 using the GR b-value). Both estimates are significantly larger than the actually observed maximum magnitude of M = 3.6 and fall outside of the 95% confidence level (see Supplementary Fig. S4) This is untypical for seismicity characterized by the GR-statistic (i.e. for an infinite seismo-tectonic continuum)[42]. It is another indication that the Groningen seismicity is entirely induced, the gas field is inherently stable and the physical conditions to trigger larger tectonic earthquakes likely do not exist.

In this paper, we presented an advanced method to investigate if an upper magnitude limit for seismicity related to gas production in the Groningen gas field exists. In contrast to purely statistical analyses, we considered information about reservoir and rupture geometries in our study. Fault ruptures seem to be restricted to the pressure perturbed reservoir layer and observed maximum magnitudes throughout the field correlate with the local reservoir thickness. Seismicity in Groningen can be characterized as exclusively induced. In this case, we find an upper induced magnitude limit of M = 4. The physical conditions for triggered earthquakes, that is, triggering of larger tectonic earthquakes on faults extending into the basement, likely do not exist. Due to the long history of production in Groningen, earthquakes of M≥4 must have occurred there several times, if possible, in disagreement with the observations. Physical reasons why tectonic seismicity is not triggered, although downgoing faults into the basement exist, might be stress barriers between the reservoir and the underlying carboniferous rocks[45,46]. Our methodology is applicable in other regions and can help to inform future regulatory actions for safer application of energy technologies.

## Methods

### FM distributions and Lower-Bound for seismicity inside a layer

The frequency magnitude (FM) distribution of seismicity in a particular region usually follows the GR law[40] describing the number of earthquakes with magnitudes ≥$M$:

$$\log_{10} N_{\geq M} = a - bM, \quad (3)$$

where a and b are typically referred to a- and b-value of the GR statistic. They describe the overall seismic activity for a given region and a given time period and the distribution between large and small magnitudes. For typical tectonic settings, the b-value is close to one. Effectively, the GR distribution describes seismicity in an infinite seismo-tectonic continuum ideally representing a given tectonic region. The infinite character of the continuum is visible from the fact that the GR statistic does not have any limitation with respect to arbitrarily large earthquakes.

FM distributions for induced seismicity (both injection and extraction induced) often differ from GR statistics. Usually, large magnitude events are underrepresented and b-values can significantly differ from one[21,28,47]. The absence of large-magnitude events can be explained by the fact that induced seismicity should only occur within a finite stimulated volume. Several approximations for the geometry of the stimulated volume exist. They usually consider an ellipsoidal volume[21,28]. Here, we use an equation, that approximates the stimulated volume as a single plane horizontal (reservoir) layer. We neglect the geometric effects of non-planar layer limitations. This is a reasonable approximation for a stimulated volume corresponding to the Groningen gas reservoir with a lateral dimension of 50km and a thickness below 0.5km (i.e., about 100 times smaller). This assumption is equivalent to an approximation of the reservoir layer by an infinite plane layer. Also, we assume that earthquakes occur on critically oriented faults. Finally, we assume that the length of the rupture is approximately proportional to its height and does not exceed its height significantly. Then the FM-distribution can be found by a magnitude integration of a product of the probability density of the GR- statistic and the probability of a potential rupture surface to be entirely within the stimulated volume[39]. This probability can be expressed in terms of the moment magnitude as $1 - 10^{\frac{M-M_Y}{2}}$, where $M_Y$ is the maximum possible induced moment magnitude. For deriving this probability we use the moment magnitude relation to the rupture length and a statistically representative (averaged) static stress drop[39] of the type of our equation (1) (main text). Finally, we obtain the number of induced earthquakes of magnitudes larger than $M$:

$$\log_{10} N_{>M} = a - bM + \log_{10}\left(1 + \frac{1}{2b-1}10^{b(M-M_Y)} - \frac{2b}{2b-1}10^{\frac{M-M_Y}{2}}\right). \quad (4)$$

In the limit of infinite $M_Y$ or very small events (large negative $M$), this frequency-magnitude distribution coincides with the GR one. Initial geometric assumptions do not explicitly affect equation (4). Thus, this equation can be generally applied to seismicity observations to search for indications of the finiteness of corresponding seismo-tectonic domains. Since b and $M_Y$ are coupled in Eq. (4) their estimation might become inaccurate if statistics are poor. Replacing the last term with the leading term in its Taylor-series expansion around $b = 1$ yields an approximation with uncoupled quantities b and $M_Y$:

$$\log_{10} N = a - bM + 2\log_{10}\left(1 - 10^{\frac{M-M_Y}{2}}\right). \quad (5)$$

Under realistic conditions, this is a very good approximation of distribution (4) (see supplementary material).

### Spatial LB and SI parameter mapping

For mapping LB-parameters spatially, we select for each location within the field those earthquakes that occurred within a 15 km radius

around this location. Subsequently, we fit Equation (5) to this FM-distribution. Additionally, we compute co-variance matrices to estimate the accuracy of the fit. If the accuracy of $M_Y$ is less than 0.2 (i.e. no clear lower-bound is visible in the data, most probably due to a poor statistic) we use the well-constrained a and b-values from the fit and neglect $M_Y$ estimates.

## Worst-Case Exceedance Probability (WCEP)

To address the possibility of triggering a large tectonic event we apply the approach recently proposed for fluid injections[31] to monitor the worst-case triggering probability in real-time. We extend this approach to fluid extraction. The original idea is to use parameters of the frequency-magnitude distribution of the infinite seismo-tectonic continuum to estimate the exceedance probability of a given (large) magnitude earthquake for the worst case. In the worst case, the seismo-tectonic continuum is characterized by the upper bound of the seismogenic index and the lower bound of the *b*-value. In the case of a fluid injection this Worst-Case Exceedance Probability (WCEP) reads[31]:

$$W_{M \geq M_{max}}(t) = 1 - \exp\left[-\Delta V_f 10^{\sup \Sigma_0(t) - \inf b(t) M_{max}}\right], \quad (6)$$

where $\Delta V_f$ is the cumulative injected fluid volume, b(t) is the GR b-value and $\Sigma_0(t)$ is the seismogenic index[29,30], which can be estimated using the GR a-value:

$$\Sigma_0(t) = a(t) - \log_{10}(\Delta V_f(t)). \quad (7)$$

In equation (6), *sup* and *inf* denote the supremum and the infimum of these values, respectively.

Here, we extend the representation of Eq. (6) to the case of a fluid production:

$$W_{M \geq M_{max}}(t) = 1 - \exp\left[-10^{\delta\Sigma(t)} 10^{\sup \Sigma_0(t) - \inf b(t) M_{max}}\right]. \quad (8)$$

The time-dependent seismogenic index can be estimated as follows

$$\Sigma_0(t) = a(t) - \delta\Sigma(t). \quad (9)$$

Generally $\delta\Sigma(t)$ is given by a normalized volumetric integration of the monotonic supremum (minimum monotonic majorant) of the Failure Coulomb Stress perturbation ($\delta FCS$) in the stimulated domain[30,44]. This can be explicitly written in the following form applicable to real event catalogues:

$$\Sigma_0(t) = \log_{10}(N_{M \geq M_c}) + M_C b(t) - \delta\Sigma(t), \quad (10)$$

where $N_M$ is the number of observed earthquakes above the magnitude of completeness $M_C$. Due to the poroelastic coupling, fluid production from a normal-faulting Groningen type of reservoir reduces the absolute value of horizontal stresses[30]. On the one hand, this leads to a stabilization of the normal-faulting regime. On the other hand, the decreasing pore pressure causes an increase of the differential stress. This increase of the differential stress represents the change of the FCS and, it is proportional to the decrease of the pore pressure[30]: $\delta FCS(t) = (\sin\phi_f(1 - n_s) - n_s)\delta Pp(t)$, where $n_s$ is the poroelastic stress-coupling coefficient. The quantity $\delta\Sigma(t)$ is then given by the logarithm of the spatial integral of $\delta FCS(t)$ over the stimulated domain with a factor normalizing to friction angle and a factor accounting for the poroelastic storage of the reservoir[30,44]. Assuming a large-area thin-layer approximation for the Groningen

reservoir, we obtain the following explicit result:

$$\delta\Sigma(t) = \log_{10}\left(AHS\left[(1 - n_s) - \frac{n_s}{\sin\phi_f}\right]\delta P_p(t)\right). \quad (11)$$

It can be split into a time-independent, field-site specific term that includes the total area of the reservoir A, the reservoir thickness H, the uni-axial storage coefficient S, the friction angle $\phi_f$ and the poroelastic stress-coupling coefficient $n_s$, and into a term that includes the temporal change of pore-pressure:

$$\delta\Sigma(t) = \log_{10}\left(AHS\left[(1 - n_s) - \frac{n_s}{\sin\phi_f}\right]\right) + \log_{10}\left[\delta P_p(t)\right]. \quad (12)$$

We use parameters $n_s = 0.375$ and $S = 5 \cdot 10^{-10}$ proposed for Groningen[30]. These estimates are consistent with the gas-extraction induced subsidence observed in Groningen[48]. Let us investigate Eq. (8) in more detail: The term $\delta\Sigma(t)$ is monotonically increasing with time. The same holds true for $\sup\Sigma_0(t)$ as we take the supremum of $\Sigma_0(t)$. $\inf b(t)$ is monotonically decreasing as we take the infimum of b. Altogether, $W_{M \geq M_{max}}(t)$ is also monotonically increasing and remains constant after the cessation of production and end of seismicity.

## Data availability

The production volumes, pore pressure and the earthquake catalog used in this study are published[49] and available under the accession code https://public.yoda.uu.nl/geo/UU01/RHHRPY.html. (https://doi.org/10.24416/UU01-RHHRPY).

## Code availability

We do not use any custom software. Pdf-prints of jupyter-notebooks (Python) are provided as Supplementary material.

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

## Acknowledgements

We thank the sponsors of the PHASE-AC consortium research project of Freie Universitaet Berlin for supporting the research presented in this paper. We thank Steve Oates and Jan van Elk for their valuable discussion and comments.

## Author contributions

S.A.S. designed the research; N.B. conducted the research; N.B., C.L. and S.A.S. analyzed the results and wrote the paper.

## Funding

## Competing interests

The authors declare no competing interests.

## Additional information

**Peer review information** : *Nature Communications* thanks Julian Bommer and Ryan Schultz for their contribution to the peer review of this work. A peer review file is available.

