## [Peer Review File · Nature Communications]

Production-Induced Seismicity Indicates a Low Risk of Strong Earthquakes in the Groningen Gas FieldREVIEWER COMMENTS

Reviewer #1 (Remarks to the Author):

This is a very interesting paper that addresses the seismicity induced by gas production from the Groningen field in the Netherlands. The authors apply a novel approach to assessing the largest magnitude of earthquake that could occur as a result of the reservoir compaction that results from the pressure depletion, but the most interesting and valuable finding is the implications for the possibility of triggered, rather than induced, earthquakes to occur. Triggered earthquakes would require fault ruptures that propagate beyond the gas-bearing formation. The authors demonstrate, through their modelling of the seismicity rates and the theoretical upper bounds on the magnitudes that could be reached, that earthquakes should already have been observed that significantly exceed the size of the largest event that has occurred in Groningen. The authors conclude, therefore, that triggered earthquakes are very unlikely to occur, a finding with potentially far-reaching consequences given that a key argument supporting the decision to shut the gas field is the possibility of larger earthquakes occurring should production continue. The approach developed by the authors could also be applied to other situations of induced seismicity and could become a tool for distinguishing cases of induced and triggered seismicity, a distinction that has received much less attention than that between induced and natural seismicity, but which is perhaps equally, if not even more, important. I believe that the manuscript merits publication in Nature Communications and I am happy to recommend that it should be accepted, but I think that the authors could improve their paper in terms of presentation. I would therefore invite the authors to consider the comments and suggestions below in preparing the final version of their manuscript. I provide these comments in the order in which they refer to the manuscript rather than any hierarchy of importance.

Line 31: Is reference appropriate for this statement?

Line 38: The moment magnitude of the 2012 Huizinge earthquakes was Mw 3.5, the value of 3.6 was the local magnitude, ML, reported by KNMI (see Dost et al., 2018, Seismological Research Letters, v.89, n.3, pp.1062-1074, and Erratum, SRL, 2019, v.90, n.4, pp.1660-1662).

Lines 38-39: It is my understanding that SodM is only an advisory body and the decision to lower gas production came from the Ministry of Economic Affairs (EZ).

Line 48: Is this Mw or ML?

Line 49: I suggest changing "stop" to something like "cessation".

Line 51: Perhaps "often" rather than "tend"?

Line 55: Are references 7-9 appropriate here?

Lines 56-72: With regards to the distinction between induced and triggered earthquakes, the authors

might consider referring to McGarr & Majer (2023), *Geothermics*, v.107, 102612.

Line 74: Delete redundant (and confusing) comma after “both”.

Line 84: Insert comma after “then”.

Line 88: Consider changing “must” to “should”.

Line 89: Insert “to” before “have physical”.

Line 94: Given the limited thickness of the reservoir, would the larger earthquakes not be expected to occur on ruptures with rather large length-to-width aspect ratios?

Line 96: Change to “(SI units are used)”

Lines 97-98: Are the dips of the faults in the field not known?

Line 98: This is a strange way to refer to the dip angle, which is conventionally measured from the horizontal. Adopting this usual convention would then change Eq.(2) to $h/\sin(\pi/3)$.

Lines 101-102: Are the dynamic stress parameters used in ground-motion modelling equivalent to the static stress drops the authors invoke here?

Line 110: Another relevant reference regarding accurate earthquake locations in Groningen might be Spetzler & Dost (2017), *Geophysical Journal International*, v.209, v.1, pp.453-465.

Line 113: Should “further” be “henceforth”? If so, the abbreviation should then be used consistently throughout the remainder of the manuscript, which is not currently the case.

Line 116: Same comment.

Lines 119-120: Consider changing to “...induced, the rupture surfaces of which are entirely...”

Line 129: Colon after “Eqs” does not look right.

Lines 139-140: The structure of this sentence is strange, please consider modifying.

Line 144: At this point, FM has not been defined.

Line 146: Insert space between comma and “b”

Line 148: Redundant to have “around” and the “ \approx ” sign (and strange to then report a magnitude value to two decimal places!)

Line 158: Consider changing “rather constant” to “that are relatively stable” or similar.

Line 162: Neither GRS nor LBS have been defined.

Line 164: The moment magnitude of the 2006 earthquake was 3.4, and ML was 3.5 (see also comment on line 38).

Lines 168-170: This sentence is very difficult to understand and needs to be expressed more clearly.

Line 173: Consider changing “get better” to “improve”.

Line 173: Delete redundant comma after “here”.

Line 192: “negligible” (without “small”) or “negligibly small”.

Line 193: “index are on the”.

Line 219: Change to “probability of triggering”.

Line 220: Change to “ruptures propagating beyond”.

Line 221: I think propagating into the basement is the only option because the Zechstein salt layer above the reservoir is not capable of supporting seismogenic rupture.

Line 225: Is the plural of index not indices?

Line 227: Suggested change: “...if high: we estimate a....”.

Line 233: Poisson with capital P.

Line 235: “expected to have happened already”.

Line 237: Ground motions have return periods, earthquakes of a given magnitude have recurrence intervals.

Line 238: same comment.

Line 238: Consider changing “never” to “not”.

Line 241: Change “by” to “with” after “volume”.

Line 242: Delete redundant bracket at the end of the line.

Line 246: Delete redundant comma after “indication”.

Lines 247-249: Suggested re-wording: “Both approaches agree in the sense that if critically stressed faults capable of producing an earthquake of $M_w \geq 4.0$ were present in Groningen, such an event would most likely have already occurred”.

Line 249: “chance of triggering large-magnitude”.

Line 254: Another relevant reference here may be Das & Scholz (1983), *Nature*, v.305, n.5935, pp.621-623.

Line 286: Delete redundant comma after “fact”.

Line 278: Use initials FM here.

Line 297: Delete redundant comma after “earthquakes”.

Line 300: Should this be “well-constrained”?

Line 305: Consider changing “productions” to “extraction”.

Eq.(12): In last term, use square and round brackets.

Line 328: Change first “stop” to “cessation” and second “stop” to “end”.

Line 348: Lacq with upper case L.

Line 357: Check authorship.

Line 410: Add full publication details (v.94, n.3, pp.1429–1446).

Figure 1 caption: Mc needs subscript (or lower case ‘c’). Also, consider removing final three zeroes from map axes numbering and inserting “x1000” in brackets after “Easting” and “Northing” to allow maps to be larger. Also state magnitude scale in (c).

Figure 4 caption: “indices” (twice).

Reviewer #2 (Remarks to the Author):

Review of *Nature Communications* article MS# NCOMMS-23-31398-T,

The manuscript of Boitz et al., “Production-Induced Seismicity Indicates a Low Risk of a Strong Earthquake in the Groningen Gas Field” is an article concerning the production-induced earthquakes at the Groningen. The authors use statistical and seismic methods to place constraints on the maximum magnitude of events that could occur there. They further suggest that there is the effect of a maximum magnitude and speculate on its cause/impacts. The results of the paper support the conclusions of the authors. I feel that the results of the paper could be interesting the readership of Nat. Comm.

That said, there are additions/revisions that are needed to improve this paper. As it currently stands, the paper could use improvements on its logical flow, organization, supporting its arguments, and additional analysis. A more detailed list of my thoughts follows below.

1. The organization of the paper could use quite a bit of work.
 - a. For example, the authors talk about equations that aren't yet introduced (Line 123). I understand that this is a high-impact, letter-style journal – so these points have to be concise. To balance these competing factors, what the authors can do is introduce the relevant equations here in a single sentence. Then they can refer the reader to the methods section for elaboration/justifications.
 - b. The authors should make use of supplementary materials. Especially for points like sensitivity tests (Figures 5 & 6 and lines 422+). These things are important to get a skeptical reader on board, to show the work has been exhaustive. However, they detract from the main story and could be simply mentioned in a single sentence in the maintext (with a reference to the supplements). The supplements can then elaborate in full detail.
2. What exactly is shown on the color map of Figure 1c? Is this the maximum magnitudes of the catalogue interpreted over space, or some average magnitude?
3. The paper makes large/grandiose claims as if they were self-evident in several spots. I'd suggest the authors use more nuance in their language. The paper also seems to lack a

series of references that could help with this point. I've listed some of the ones that stood out to me below. I'd suggest that the authors go back and do a more thorough literature search, as they've only used 37 out of the 70 references that this journal allows.

- a. Introduction: there's a good overview paper on the Groningen (and Dutch earthquakes in general) that should be cited [Muntendam-Bos et al., 2022].
Muntendam-Bos, A. G., Hoedeman, G., Polychronopoulou, K., Draganov, D., Weemstra, C., van der Zee, W., ... & Roest, H. (2022). An overview of induced seismicity in the Netherlands. *Netherlands Journal of Geosciences*, 101, e1.
 - b. This statement "maximum induced earthquake magnitudes tend to occur after termination of energy projects" is a bit of a stretch. Sometimes big earthquakes have happened after – not that they tend to. There's a nice paper that quantifies exactly this problem [Schultz et al., 2022].
Schultz, R., Ellsworth, W. L., & Beroza, G. C. (2022). Statistical bounds on how induced seismicity stops. *Scientific reports*, 12(1), 1184.
 - c. Line 73 "all approaches agree that induced seismicity is much easier to control than..." care to back this claim up with a reference?
4. Line 54 has a bit of awkward wording where it seems to be making the distinction between triggered and stimulated events, but the next paragraph more correctly makes the distinction between triggered and induced events.
 5. The authors make a series of assumptions about the largest possible fault plane: that the events are truly induced, the reservoir height provides a boundary, that the reactivated faults are critically oriented (for the normal stress field), that strike-dip lengths are equal. I'd like to see more emphasis placed on justifying and discussing these assumptions.
 - a. In reality, earthquakes won't be perfectly induced or just triggered. They will be somewhere in between. How well does using the reservoir boundaries actually hold?
 - b. There doesn't appear to be any real reason why the strike-dip lengths of the events need to be the same. In large ruptures (like along San Andreas), it's known that these rupture geometries can be different.

- c. Why should the critical orientation hold? Groningen went on for decades, and significantly altered the stress state. Presumably faults that were sub-optimal also were reactivated here.
 - d. Can the authors compare against either the 3D reflection seismic data for Groningen or the hypocentre geometries to assess how well these assumptions apply? This would be the easiest way to assert/refute these assumptions. For example, do the earthquakes all occur within the reservoir, or is there some leakage into surrounding strata? Are all of the hypocentres along optimally oriented faults? This would also be an opportunity for more papers to cite and discuss.
6. How much better is the truncated distribution fit than the regular fit? Is it better enough to justify the use of an extra free parameter? Just by introducing extra fit parameters, we would automatically expect the fit to be better.
7. Line 179: we could also think about discerning between these two scenarios by biasing in the seismogenic index fit. When there's a temporal delay introduced between an operation and the responding seismicity, then seismogenic index would be expected to start low and then increase towards its true value.
8. The results showing a rough correspondence of the MY with the reservoir height is really cool! The authors should do a bit more work here: quantify how well their models and predictions fit together. For example, they could show a scatter plot between the MY modelled from reservoir height (Eqns 1 & 2) and the catalogue measured MY for each location. Does this information actually predict what's measured?
9. The following page contains a list of minor mistakes.

Thanks,

-Ryan

The abstract has a nice symbol for Mw and then a text version elsewhere.

Line 22: This sentence is awkwardly phrased, maybe say “Due to the long history of production in Groningen, *our model estimates that* earthquakes of $M_w \geq 4$ would have occurred there several times, in disagreement with the observations”. Similar statement for line 88.

Line 24: this might not be the right thing to say here either.

Sneak in a citation to some of my works? Maybe HF-IS or TLPnl?

Figure 1 caption: “MC” formatting should be “MC”.

General: why is this a lowerbound model? Aren't we placing a cap on what the upperbound on what the maximum magnitude of an earthquake can be?

Line 136: fitting a catalogue to the LB-statistic doesn't get the seismogenic index. Also the phrasing here is awkward. It took me a couple of tries to figure out the authors meant that there is a finite and infinite versions of the parameters.

Line 224: “indexes” should be “indicies”

Line 254: The paper just feels like it suddenly ends.

Figure 3: panels b & c seem inversely related.

Line 281: this isn't exactly obvious that this formulation should hold.

Line 433: “statistical error” should be “statistical variance”

I'm also curious to the authors thoughts on how their results might impact induced seismicity management plans, like traffic lights. Having an accurate estimate of maximum magnitudes would be quite important for how these are designed [Schultz et al., 2022] – both for Dutch gas fields, but also future industries.

Schultz, R., Muntendam-Bos, A., Zhou, W., Beroza, G. C., & Ellsworth, W. L. (2022). Induced seismicity red-light thresholds for enhanced geothermal prospects in the Netherlands. *Geothermics*, 106, 102580.

REVIEWER COMMENTS

Reviewer #1 (Remarks to the Author):

This is a very interesting paper that addresses the seismicity induced by gas production from the Groningen field in the Netherlands. The authors apply a novel approach to assessing the largest magnitude of earthquake that could occur as a result of the reservoir compaction that results from the pressure depletion, but the most interesting and valuable finding is the implications for the possibility of triggered, rather than induced, earthquakes to occur. Triggered earthquakes would require fault ruptures that propagate beyond the gas-bearing formation. The authors demonstrate, through their modelling of the seismicity rates and the theoretical upper bounds on the magnitudes that could be reached, that earthquakes should already have been observed that significantly exceed the size of the largest event that has occurred in Groningen. The authors conclude, therefore, that triggered earthquakes are very unlikely to occur, a finding with potentially far-reaching consequences given that a key argument supporting the decision to shut the gas field is the possibility of larger earthquakes occurring should production continue. The approach developed by the authors could also be applied to other situations of induced seismicity and could become a tool for distinguishing cases of induced and triggered seismicity, a distinction that has received much less attention than that between induced and natural seismicity, but which is perhaps equally, if not even more, important. I believe that the manuscript merits publication in Nature Communications and I am happy to recommend that it should be accepted, but I think that the authors could improve their paper in terms of presentation. I would therefore invite the authors to consider the comments and suggestions below in preparing the final version of their manuscript. I provide these comments in the order in which they refer to the manuscript rather than any hierarchy of importance.

We agree that the presentation of the paper needed to be improved. In the revised version of the manuscript, we have introduced a clearer organization to make sure the main part of the paper stands on its own and can be read and understood without the need to read additional references and the Supplementary Information. We moved details to the Supplementary Information where we elaborate on some points in full detail. The manuscript is now clearly separated into the main article including the method section and the Supplementary information as separate files.

We also introduced a Conclusions section to make our main points and findings more accessible and clearer. Finally, we worked on substantiating our claims and findings by references and additional statistical analysis. For details see the revised manuscript with marked changes.

Line 31: Is reference appropriate for this statement?

Not really, it has been removed here. Additionally, we added some references of papers published after the workshop (Line 50) that discuss different M_{max} estimates.

Line 38: The moment magnitude of the 2012 Huizinge earthquakes was M_w 3.5, the value of 3.6 was the local magnitude, M_L , reported by KNMI (see Dost et al., 2018, Seismological Research Letters, v.89, n.3, pp.1062-1074, and Erratum, SRL, 2019, v.90, n.4, pp.1660-1662).

We added both estimates in the text, as well as the Dost et al. 2018 Reference

Lines 38-39: It is my understanding that SodM is only an advisory body and the decision to lower gas production came from the Ministry of Economic Affairs (EZ).

We changed the sentence to “After the largest earthquake (Mw 3.5, ML 3.6) in 2012 lower production rates were mandated in the Groningen field.

Line 48: Is this Mw or ML?

These are the magnitudes from the published KNMI catalog which provides magnitudes as M_L. Therefore, we changed the Mw to ML. However, one of the results of the Dost et al. 2018 is that particular for large magnitude events the difference between ML and Mw is negligible.

Line 49: I suggest changing “stop” to something like “cessation”.

Changed

Line 51: Perhaps “often” rather than “tend”?

Changed

Line 55: Are references 7-9 appropriate here?

No, they have been removed

Lines 56-72: With regards to the distinction between induced and triggered earthquakes, the authors might consider referring to McGarr & Majer (2023), *Geothermics*, v.107, 102612.

We added the reference in line 56 of the plain revised manuscript

Line 74: Delete redundant (and confusing) comma after “both”.

Changed

Line 84: Insert comma after “then”.

Changed

Line 88: Consider changing “must” to “should”.

Changed

Line 89: Insert “to” before “have physical”.

Changed

Line 94: Given the limited thickness of the reservoir, would the larger earthquakes not be expected to occur on ruptures with rather large length-to-width aspect ratios?

By deriving the LB statistic a simplifying assumption about a proportionality of strike and dip lengths of ruptures is accepted initially. However, the resulting equations contain the maximum possible magnitude parameter M_V only. On the one hand this parameter has a nature of an effective limiting magnitude representing a statistical set of various rupture geometries. On the other hand, realistically the strike and dip lengths of ruptures are usually of the same order of scale. Indeed, our rough estimates of the maximum magnitude made using the height of the reservoir in Groningen are in a well agreement with the observations. Also the fit of the LB statistic to the data shows a close value, supporting thus this assumption again.

Line 96: Change to “(SI units are used)”

Changed

Lines 97-98: Are the dips of the faults in the field not known?

Fault dips from moment tensor solutions are in the range of 50-70° (Willacy et al, 2019 and Dost et al. 2020) and thus our 60° a reasonable assumption. We added this information and both references to the manuscript.

Line 98: This is a strange way to refer to the dip angle, which is conventionally measured from the horizontal. Adopting this usual convention would then change Eq.(2) to $h/\sin(\pi/3)$.

We adopted the usual convention and changed the text and equation correspondingly.

Lines 101-102: Are the dynamic stress parameters used in ground-motion modelling equivalent to the static stress drops the authors invoke here?

We assume that they are at least of the same order of magnitude, the stress drops estimated from the Brune model (it seems this is used for ground-motion) and the static stress drop used in the Magnitude-Length relation.

Line 110: Another relevant reference regarding accurate earthquake locations in Groningen might be Spetzler & Dost (2017), *Geophysical Journal International*, v.209, v.1, pp.453-465.

We added the reference and “the majority of hypocenter locations ...” in the text.

Line 113: Should “further” be “henceforth”? If so, the abbreviation should then be used consistently throughout the remainder of the manuscript, which is not currently the case.

Yes. We changed further to henceforth and use LB for the remainder of the manuscript

Line 116: Same comment.

Abbreviation GR is used for the remainder of the manuscript

Lines 119-120: Consider changing to “...induced, the rupture surfaces of which are entirely...”

Changed

Line 129: Colon after “Eqs” does not look right.

Removed

Lines 139-140: The structure of this sentence is strange, please consider modifying.

We have rephrased the whole paragraph

Line 144: At this point, FM has not been defined.

Text is changed to “frequency-magnitude (FM) -distribution”

Line 146: Insert space between comma and “b”

Changed

Line 148: Redundant to have “around” and the “≈” sign (and strange to then report a magnitude value to two decimal places!)

Correct, we removed “around” and the “≈” sign and added ‘equal’ as this is the result of the fitting (with the uncertainty given in Figure 2c)

Line 158: Consider changing “rather constant” to “that are relatively stable” or similar.

Modified as proposed

Line 162: Neither GRS nor LBS have been defined.

Those abbreviations have been removed from the manuscript. We now use consistently “LB-statistic” and “GR-statistic”

Line 164: The moment magnitude of the 2006 earthquake was 3.4, and ML was 3.5 (see also comment on line 38).

Text changed to ML

Lines 168-170: This sentence is very difficult to understand and needs to be expressed more clearly.

We separated this sentence into two: “The two large-magnitude events in 2006 and 2012 caused an increase in both the maximum-magnitude estimate and its uncertainty. In the subsequent years these estimates decreased (Figure 2f).”

Line 173: Consider changing “get better” to “improve”.

Line 171? We changed the text as proposed

Line 173: Delete redundant comma after “here”.

Changed

Line 192: “negligible” (without “small”) or “negligibly small”.

Changed to “negligibly small”

Line 193: “index are on the”.

Changed

Line 219: Change to “probability of triggering”.

Changed

Line 220: Change to “ruptures propagating beyond”.

Changed

Line 221: I think propagating into the basement is the only option because the Zechstein salt layer above the reservoir is not capable of supporting seismogenic rupture.

Very correct. There is a thin layer of anhydrite (?) above, where some rupture should be possible.

Line 225: Is the plural of index not indices?

We now use consistently indices throughout the manuscript.

Line 227: Suggested change: "...if high: we estimate a....".

Changed to "...is high: we estimate a...."

Line 233: Poisson with capital P.

Changed

Line 235: "expected to have happened already".

Changed

Line 237: Ground motions have return periods, earthquakes of a given magnitude have recurrence intervals.

Modified

Line 238: same comment.

Modified

Line 238: Consider changing "never" to "not".

Changed

Line 241: Change "by" to "with" after "volume".

Changed

Line 242: Delete redundant bracket at the end of the line.

Changed

Line 246: Delete redundant comma after "indication".

Removed

Lines 247-249: Suggested re-wording: "Both approaches agree in the sense that if critically stressed faults capable of producing an earthquake of $M_w \geq 4.0$ were present in Groningen, such an event would most likely have already occurred".

Changed as proposed

Line 249: "chance of triggering large-magnitude".

Changed

Line 254: Another relevant reference here may be Das & Scholz (1983), Nature, v.305, n.5935, pp.621-623.

Reference added

Line 286: Delete redundant comma after "fact".

Removed

Line 278: Use initials FM here.

Changed, as well as GR-statistics in 279

Line 297: Delete redundant comma after "earthquakes".

Removed

Line 300: Should this be "well-constrained"?

Yes, we changed that

Line 305: Consider changing "productions" to "extraction".

Changed

Eq.(12): In last term, use square and round brackets.

Changed

Line 328: Change first "stop" to "cessation" and second "stop" to "end".

Changed

Line 348: Lacq with upper case L.

Changed

Line 357: Check authorship.

Corrected

Line 410: Add full publication details (v.94, n.3, pp.1429–1446).

Added

Figure 1 caption: Mc needs subscript (or lower case 'c'). Also, consider removing final three zeroes from map axes numbering and inserting "x1000" in brackets after "Easting" and "Northing" to allow maps to be larger. Also state magnitude scale in (c).

Mc has now a subscript c, Figure 1c) was updated with magnitude scale ML. For the maps we would like to keep the current notation.

Figure 4 caption: "indices" (twice).

As stated above we use consistently "indices"

Review of Nature Communications article MS# NCOMMS-23-31398-T,

The manuscript of Boitz et al., “Production-Induced Seismicity Indicates a Low Risk of a Strong Earthquake in the Groningen Gas Field” is an article concerning the production-induced earthquakes at the Groningen. The authors use statistical and seismic methods to place constraints on the maximum magnitude of events that could occur there. They further suggest that there is the effect of a maximum magnitude and speculate on its cause/impacts. The results of the paper support the conclusions of the authors. I feel that the results of the paper could be interesting the readership of Nat. Comm.

That said, there are additions/revisions that are needed to improve this paper. As it currently stands, the paper could use improvements on its logical flow, organization, supporting its arguments, and additional analysis. A more detailed list of my thoughts follows below.

The organization of the paper could use quite a bit of work.

We agree that the organization of the paper needed to be improve. In the revised version of the manuscript, we have introduced a clearer organization to make sure the main part of the paper stands on its own and can be read and understood without the need to read additional references and the Supplementary Information. We moved details to the Supplementary Information where we elaborate on some points in full detail. The manuscript is now clearly separated into the main article including the method section and the Supplementary information as separate files.

We also introduced a Conclusions section to make our main points and findings more accessible and clearer. Finally, we worked on substantiating our claims and findings by references and additional statistical analysis. For details see the revised manuscript with marked changes.

For example, the authors talk about equations that aren't yet introduced (Line 123). I understand that this is a high-impact, letter-style journal – so these points have to be concise. To balance these competing factors, what the authors can do is introduce the relevant equations here in a single sentence. Then they can refer the reader to the methods section for elaboration/justifications.

During the revision of the manuscript, we made sure to introduce all the relevant equations before discussing them in detail. We now clearly refer to the method section where needed for elaboration and justifications.

See e.g. the mentioned Line 123 in the originally submitted article (introduction of the Lower-Bound equations):

Figure 2a shows a sketch of our application of the LB-concept to a single stimulated horizontal layer. Blue lines indicate possible, and red lines impossible rupture surfaces according to the LB-statistic. The dashed blue line indicates the largest possible rupture surface. Using these limiting conditions for possible rupture sizes, we derive explicit equations of the Lower-Bound frequency-magnitude statistics of induced seismicity (see Methods, Eqs. 4 and 5).

The authors should make use of supplementary materials. Especially for points like sensitivity tests (Figures 5 & 6 and lines 422+). These things are important to get a skeptical reader on board, to show the work has been exhaustive. However, they detract from the main story and could be simply mentioned in a single sentence in the maintext (with a reference to the supplements). The supplements can then elaborate in full detail.

The manuscript is now clearly separated into the main article including the Method section and the Supplementary information as separate files. It was our intention to have the mentioned sections as a Supplementary Information. Unfortunately, it was not clear. We have now split the documents into the main manuscript (everything before line 422 in the initial manuscript) and moved the remaining part to a Supplementary Information file. In the main manuscript we refer to this supplementary file where necessary.

2. What exactly is shown on the color map of Figure 1c? Is this the maximum magnitudes of the catalogue interpreted over space, or some average magnitude?

You either refer here to Fig. 1b or 1d? For 1b we compute the maximum magnitude for reservoir limited ruptures (Eq. 1 in the revised manuscript) using the known reservoir thickness and a constant stress drop of 10MPa. For this plot, catalogue magnitudes are not used. For Fig. 1d we plot the **maximum** magnitudes from the event catalogue interpreted over space. We added “maximum observed” in the caption of the Figure.

The paper makes large/grandiose claims as if they were self-evident in several spots. I'd suggest the authors use more nuance in their language. The paper also seems to lack a series of references that could help with this point. I've listed some of the ones that stood out to me below. I'd suggest that the authors go back and do a more thorough literature search, as they've only used 37 out of the 70 references that this journal allows.

We understand that some of our claims needed to be supported by additional analysis and references. We worked on substantiating our claims and findings by references and additional statistical analysis. Also, we worked on the language and structure of the manuscript to support our claims. For details see the revised manuscript with marked changes. We also added the suggested references and additional relevant references based on a literature search.

Introduction: there's a good overview paper on the Groningen (and Dutch earthquakes in general) that should be cited [Muntendam-Bos et al., 2022].

Muntendam-Bos, A. G., Hoedeman, G., Polychronopoulou, K., Draganov, D., Weemstra, C., van der Zee, W., ... & Roest, H. (2022). An overview of induced seismicity in the Netherlands. *Netherlands Journal of Geosciences*, 101, e1.

Thank you for the suggestion. This is a relevant reference and we added it in the revised manuscript.

This statement "maximum induced earthquake magnitudes tend to occur after termination of energy projects" is a bit of a stretch. Sometimes big earthquakes have happened after – not that they tend to. There's a nice paper that quantifies exactly this problem [Schultz et al., 2022].

Schultz, R., Ellsworth, W. L., & Beroza, G. C. (2022). Statistical bounds on how induced seismicity stops. *Scientific reports*, 12(1), 1184.

Changed to maximum induced earthquake magnitudes sometimes occur after termination of energy projects. We added the suggested reference Schultz et al. 2022 to support this point.

c) Line 73 "all approaches agree that induced seismicity is much easier to control than..." care to back this claim up with a reference?

We added references and rephrased the sentence (see the revised manuscript):

“Induced seismicity is controlled mainly by technical operations and depends less on tectonic features. Thus, it is easier to control than triggered seismicity by modifying operation activity based on application of traffic-light systems^{18, 23, 24}.”

References added:

19. Schultz, R., Ellsworth, W. L. & Beroza, G. C. Statistical bounds on how induced seismicity stops. *Scientific reports* 12, 1184 (2022).482

23. Mignan, A., Broccardo, M., Wiemer, S. & Giardini, D. Induced seismicity closed-form traffic light system for actuarial decision-making during deep fluid injections. *Scientific reports* 7, (2017).

24. Kwiatek, G. et al. Controlling fluid-induced seismicity during a 6.1-km-deep geothermal stimulation in finland. *Science advances* 5, eaav7224 (2019).

Line 54 has a bit of awkward wording where it seems to be making the distinction between triggered and stimulated events, but the next paragraph more correctly makes the distinction between triggered and induced events.

We rephrased the sentence in the revised manuscript:

“In order to understand the nature of a maximum possible earthquake, a distinction must be made between earthquakes that are, on the one hand, predominantly caused by geotechnical interventions in the subsoil, and, on the other hand, tectonically prepared earthquakes that are triggered by anthropogenic influences^{7-9, 19-21}.” (for references see the revised manuscript)

5. The authors make a series of assumptions about the largest possible fault plane: that the events are truly induced, the reservoir height provides a boundary, that the reactivated faults are critically oriented (for the normal stress field), that strike-dip lengths are equal. I'd like to see more emphasis placed on justifying and discussing these assumptions.

We have added more discussion to support our assumptions in the revised manuscript.

“The Groningen gas field is under normal-faulting conditions and we assume that ruptures are dipping occur under 60° to the horizontal direction, in good agreement with dips derived from moment tensor solutions^{30, 31}.”

Added references:

30. Willacy, C. et al. Full-waveform event location and moment tensor inversion for induced seismicity. *GEOPHYSICS* 84, KS39–KS57 (2019).

31. Dost, B. et al. Probabilistic moment tensor inversion for hydrocarbon-induced seismicity in the groningen gas field, the netherlands, part 2: Application. *Bulletin of the Seismological Society of America* 110, 2112–2123 (2020).

Also see the revised manuscript:

“An increase of the maximum magnitude from the SE to the NW is actually observed. It is substantiated by a statistical analysis of the event-size distribution of the Groningen extraction-induced seismicity catalogue¹³ concluding that the probability of larger magnitude events in the NW-region is statistically significantly larger than in the southern and eastern parts of the gas field. In addition, the majority of observed precise hypocenter locations of seismicity in Groningen are indeed restricted to the reservoir layer^{30, 31, 35}.”

Added references:

13. Muntendam-Bos, A. G. & Grobbe, N. Data-driven spatiotemporal assessment of the event-size distribution of the groningen extraction-induced seismicity catalogue. *Scientific reports* 12, 10119 (2022)

35. Spetzler, J. & Dost, B. Hypocenter estimation of induced earthquakes in groningen. *Geophysical Journal International* ggx020 (2017)

5a) In reality, earthquakes won't be perfectly induced or just triggered. They will be somewhere in between. How well does using the reservoir boundaries actually hold?

Indeed, we assume that ruptures of the induced earthquakes restricted by boundaries of a layer of a given height. On the one hand, this agrees completely with our definition of induced earthquakes. The functional form of the LB statistic follows from this assumption. This form fits well the real-data frequency-magnitude distribution. On the other hand, 3-D distribution of hypocenters of Groningen earthquake clearly indicate their location in a restricted-height layer (Willacy et al., 2019 and Dost et al., 2020). See also general replies to point 6 above and the revised manuscript.

5b) There doesn't appear to be any real reason why the strike-dip lengths of the events need to be the same. In large ruptures (like along San Andreas), it's known that these rupture geometries can be different.

Strictly speaking by deriving the LB-statistic a simplifying assumption about a proportionality of strike and dip lengths of ruptures is accepted initially. However, the resulting equations contain the maximum possible magnitude parameter M_Y only. On the one hand this parameter has a nature of an effective limiting magnitude representing a statistical set of various rupture geometries. On the other hand, realistically the strike and dip lengths of ruptures are usually of the same order of scale. Indeed, our rough estimates of the maximum magnitude made using the height of the reservoir in Groningen are in a well agreement with the observations. Also, the fit of the LB statistic to the data agrees with the maximum observed maximum limit, supporting thus this assumption again.

See the revised manuscript:

„Another useful feature of our LB-statistic is its independence of any (potentially unknown) geometric parameter of the stimulated volume, as all LB-statistic parameters can be obtained by fitting Eq. 4 to the observed seismicity. Thus, our main geometrical assumptions are not significantly restrictive and the LB-statistic can be generally applied to seismicity observations to search for indications of the finiteness of seismically active domains.”

5c) Why should the critical orientation hold? Groningen went on for decades, and significantly altered the stress state. Presumably faults that were sub-optimal also were reactivated here.

See point b above: Our main geometrical assumptions are not significantly restrictive. Additionally, the moment tensors solutions of the earthquakes in Groningen field^{30, 31} do not show a systematic change in orientation. Rotations of the orientations seem to be in the order of $\pm 10^\circ$ (Willacy et al., 2019, Dost et al., 2020) and thus reactivation of sub-optimal oriented rupture planes will not significantly change the limiting size of rupture surfaces.

5d) Can the authors compare against either the 3D reflection seismic data for Groningen or the hypocentre geometries to assess how well these assumptions apply? This would be the easiest way to assert/refute these assumptions. For example, do the earthquakes all occur within the

reservoir, or is there some leakage into surrounding strata? Are all of the hypocentres along optimally oriented faults? This would also be an opportunity for more papers to cite and discuss.

See replies to points a-c above and:

Again, known fault plane solutions and moment tensor inversion show indeed predominantly critical orientations of rupture surfaces. (Dost et al.,2020).

Hypocenters are inside a restricted layer (Willacy et al., 2019,Dost et al.,2020). Fault dips are between 50-70°, which is in good agreement with our assumption of 60°. Fault dips from 3D reflection data are estimated mostly between 70 and 85° (Kortekaas and Jaarsma, 2017). The long-term production of Gas in Groningen significantly depleted the pore pressure and led to a reduction of the horizontal stresses due to the poroelastic coupling. However, the reservoir remained in average in the normal stress regime. It is possible that more cohesive and sub-optimal faults were also reactivated. However, it seems that the orientation fluctuations do not have large systematic deviations. Please note also that statistical orientation deviations are also accounted for in the effective parameter M_Y to the observed magnitude frequency distribution. The assumed geometric parameters are not explicitly included in the Lower-Bound-statistic (Eq. 4 and 5) fitted to the frequency-magnitude distribution and are thus not significantly restrictive.

6. How much better is the truncated distribution fit than the regular fit? Is it better enough to justify the use of an extra free parameter? Just by introducing extra fit parameters, we would automatically expect the fit to be better.

This is indeed an important point. We investigated this using the Akaike Information Criterion. Despite an additional parameter the LB-model is characterized by a lower AIC (lower information loss) in the magnitude range $M \geq 1.5$ and thus is the preferred model. We added the AIC analysis in the Supplementary Information and the revised main part of the manuscript.

“We used the Akaike Information Criterion⁴⁰ (AIC) to assess if the GR-model or the LB-model is the preferred model to describe the observed frequency-magnitude distribution. The AIC penalizes additional parameters in a consider model to prevent over-fitting. While the fit of the LB-statistic obviously can explain the observed frequency-magnitude distribution more accurate (especially the systematic deviation for larger magnitudes) it has an additional free parameter (M_Y). Nevertheless, the LB-model is characterized by a lower AIC (lower information loss) in the magnitude range $M \geq 1.5$ and thus the preferred model (see the Supplementary Information). It is again an indication that observed seismicity in Groningen is exclusively induced and an upper

magnitude limit at $M = 4$ exists.”

Added reference:

40. Akaike, H. A new look at the statistical model identification. IEEE Trans. Autom. Control 19535 (1974).

7. Line 179: we could also think about discerning between these two scenarios by biasing in the seismogenic index fit. When there’s a temporal delay introduced between an operation and the responding seismicity, then seismogenic index would be expected to start low and then increase towards its true value.

We do not consider any systematic delay between the operation and seismicity responding, because the pore pressure was depleted more or less homogeneously in the reservoir. Our scenarios are: 1. a minimum poroelastic perturbation necessary for inducing events is nearly 0; and 2. a minimum poroelastic perturbation is significant and was achieved in 1992. We do this to estimate the effect of a potentially missing seismicity observations before 1992. Scenario 2, albeit physically realistic, is effectively indistinguishable with missing observations. Thus, we compute the SI for both scenarios to estimate a possible effect of these scenarios on the obtained values of the SI and the resulting magnitude exceedance probabilities.

8. The results showing a rough correspondence of the M_Y with the reservoir height is really cool! The authors should do a bit more work here: quantify how well their models and predictions fit together. For example, they could show a scatter plot between the M_Y modelled from reservoir height (Eqns 1 & 2) and the catalogue measured M_Y for each location. Does this information actually predict what’s measured?

The following figures show exactly such scatter plots. The left plot shows the observed M_{max} vs M_{max} estimated from the thickness (for two different stress drops of 1 and 10MPa), the middle subplot the observed M_{max} vs M_Y obtained from fitting the LB-equation and the right subplot M_{max} from the thickness vs M_Y from fitting the LB-equation.

Subplot a) shows a clear correlation between the observed magnitudes and reservoir height, Subplots b) and c) show that maximum magnitudes obtained from the lower-bound fit are significantly larger than observed and expected from the thickness. Nevertheless, this is a reasonable result as the LB provides an upperbound for the magnitude. An event of such magnitude might not have occurred yet.

Also, we have added see the revised manuscript:

“An increase of the maximum magnitude from the SE to the NW is actually observed. It is substantiated by a statistical analysis of the event-size distribution of the Groningen extraction-induced seismicity catalogue¹³ concluding that the probability of larger magnitude events in the NW-region is statistically significantly larger than in the southern and eastern parts of the gas field.”

The abstract has a nice symbol for M_w and then a text version elsewhere.

We now use consistently “ M_w ” throughout the whole paper.

Line 22: This sentence is awkwardly phrased, maybe say “Due to the long history of production in Groningen, our model estimates that earthquakes of $M_w \geq 4$ would have occurred there several times, in disagreement with the observations”. Similar statement for line 88.

We added “our model estimates that” in both cases as proposed.

Line 24: this might not be the right thing to say here either.

“The seismic hazard likely is controlled only by the magnitude-limited production-induced earthquakes”. We believe this statement is supported by the analysis in the manuscript. We find that likely seismicity is exclusively induced, and ruptures are limited to the reservoir. Thus, an upper magnitude limit likely exists and larger triggered tectonic earthquakes are unlikely.

Sneak in a citation to some of my works? Maybe HF-IS or TLPnl?

We added Schultz et al, 2022 reference in line 81 of the revised version.

Figure 1 caption: “MC” formatting should be “Mc”.

Corrected

General: why is this a lowerbound model? Aren't we placing a cap on what the upperbound on what the maximum magnitude of an earthquake can be?

The model was introduced as the “lower bound of seismicity“ (Shapiro et al, 2013) as it provides a lower limit for the frequency magnitude distribution. In the frequency magnitude plot (Fig. 2c) it falls below the classical GR-statistic and thus was defined as lower-bound. In turn this means as you understood correctly it is an upper limit for the maximum magnitude.

Line 136: fitting a catalogue to the LB-statistic doesn't get the seismogenic index. Also the phrasing here is awkward. It took me a couple of tries to figure out the authors meant that there is a finite and infinite versions of the parameters.

We worked on the structure and language of the manuscript to clarify this point. Also we now explain in more detail how the parameters are estimated and explain their meaning in more detail.

Line 224: “indexes” should be “indicies”

All “indexes” throughout the paper have been changed to “indices”

Line 254: The paper just feels like it suddenly ends.

We restructured the manuscript and introduced a Conclusions section to make our main points and findings more accessible and clearer. The paper now has a clear end after the main points are summarized in the Conclusions. For details see the revised manuscript with marked changes.

Figure 3: panels b & c seem inversely related.

These two quantities are separate fitting parameters and don't have to be related. Figure 3c shows a good correspondence with the theoretical expectations (Figure 1b) and the b-value map shows the separation into NW and SE of the field that has been also observed by others (e.g. Muntendam-Bos and Grobbe, 2022).

Line 281: this isn't exactly obvious that this formulation should hold.

We are not sure, what is unclear here.

Line 433: "statistical error" should be "statistical variance"

Corrected

I'm also curious to the authors thoughts on how their results might impact induced seismicity management plans, like traffic lights. Having an accurate estimate of maximum magnitudes would be quite important for how these are designed [Schultz et al., 2022] – both for Dutch gas fields, but also future industries.

Schultz, R., Muntendam-Bos, A., Zhou, W., Beroza, G. C., & Ellsworth, W. L. (2022). Induced seismicity red-light thresholds for enhanced geothermal prospects in the Netherlands. *Geothermics*, 106, 102580.

We added the reference and a short note on traffic-light systems in line 81 of the revised manuscript. A general application to traffic light systems is difficult since not the LB-method requires a sufficient number of observed earthquakes and not all case studies might exhibit a LB magnitude statistic.

REVIEWERS' COMMENTS

Reviewer #1 (Remarks to the Author):

Thank you for the attention paid to my original review comments. I am satisfied with the responses and the changes made to the paper. I would only note that in citing Dost et al. (2018) it is also important to cite the 2019 erratum to that paper, which I also referred to in my original review.

Reviewer #2 (Remarks to the Author):

Review of Nature Communications article MS# NCOMMS-23-31398-T,

The manuscript of Boitz et al., "Production-Induced Seismicity Indicates a Low Risk of a Strong Earthquake in the Groningen Gas Field" is an article concerning the production-induced earthquakes at the Groningen. In this revised version of the paper the authors have made significant revisions to better organize and clarify their work. I feel they have suitably addressed my comments and would recommend it for publication.

I only have a few small points to make.

- I'm glad that the authors made the scatter plots comparing various estimates M_{max} estimates. When I asked that in my review, I meant that the authors should include these results in the paper. Adding the scatter plot (shown in the rebuttal document) to the supplements would be nice for other similarly curious readers to examine.
- The sentence "Note that at no point in time the estimated maximum possible magnitude is smaller than the observation of $M_{max} = 3.6$ " is awkwardly phrased.
- Line 444 (of the track-changes document): "well-constraint" should be "well-constrained".

Thanks,
-Ryan

REVIEWERS' COMMENTS

Reviewer #1 (Remarks to the Author):

Thank you for the attention paid to my original review comments. I am satisfied with the responses and the changes made to the paper. I would only note that in citing Dost et al. (2018) it is also important to cite the 2019 erratum to that paper, which I also referred to in my original review.

We added the reference

Reviewer #2 (Remarks to the Author):

Review of Nature Communications article MS# NCOMMS-23-31398-T,

The manuscript of Boitz et al., "Production-Induced Seismicity Indicates a Low Risk of a Strong Earthquake in the Groningen Gas Field" is an article concerning the production-induced earthquakes at the Groningen. In this revised version of the paper the authors have made significant revisions to better organize and clarify their work. I feel they have suitably addressed my comments and would recommend it for publication.

I only have a few small points to make.

- I'm glad that the authors made the scatter plots comparing various estimates M_{max} estimates. When I asked that in my review, I meant that the authors should include these results in the paper. Adding the scatter plot (shown in the rebuttal document) to the supplements would be nice for other similarly curious readers to examine.

We added the figure and a short explanation in the Supplementary Material

- The sentence "Note that at no point in time the estimated maximum possible magnitude is smaller than the observation of $M_{max} = 3.6$ " is awkwardly phrased.

We rephrased the sentence: "Note that all temporal estimates of the maximum possible magnitude $M_{Y\text{\$}}$ are larger than the observed $M_{\text{\$}\{max\}=3.6\text{\$}}$."

- Line 444 (of the track-changes document): "well-constraint" should be "well-constrained".

Corrected

Thanks,
-Ryan